# Molecular Characterization of Small Ruminant Lentiviruses in Polish Mixed Flocks Supports Evidence of Cross Species Transmission, Dual Infection, a Recombination Event, and Reveals the Existence of New Subtypes within Group A

**DOI:** 10.3390/v13122529

**Published:** 2021-12-16

**Authors:** Monika Olech, Jacek Kuźmak

**Affiliations:** 1Department of Swine Diseases, National Veterinary Research Institute, 24-100 Pulawy, Poland; 2Department of Biochemistry, National Veterinary Research Institute, 24-100 Pulawy, Poland; jkuzmak@piwet.pulawy.pl

**Keywords:** small ruminant lentiviruses, SRLV, sheep, goat, subtype, recombination

## Abstract

Small ruminant lentiviruses (SRLVs) are a group of highly divergent viruses responsible for global infection in sheep and goats. In a previous study we showed that SRLV strains found in mixed flocks in Poland belonged to subtype A13 and A18, but this study was restricted only to the few flocks from Małopolska region. The present work aimed at extending earlier findings with the analysis of SRLVs in mixed flocks including larger numbers of animals and flocks from different part of Poland. On the basis of *gag* and *env* sequences, Polish SRLVs were assigned to the subtypes B2, A5, A12, and A17. Furthermore, the existence of a new subtypes, tentatively designed as A23 and A24, were described for the first time. Subtypes A5 and A17 were only found in goats, subtype A24 has been detected only in sheep while subtypes A12, A23, and B2 have been found in both sheep and goats. Co-infection with strains belonging to different subtypes was evidenced in three sheep and two goats originating from two flocks. Furthermore, three putative recombination events were identified within *gag* and *env* SRLVs sequences derived from three sheep. Amino acid (aa) sequences of immunodominant epitopes in CA protein were well conserved while Major Homology Region (MHR) had more alteration showing unique mutations in sequences of subtypes A5 and A17. In contrast, aa sequences of surface glycoprotein exhibited higher variability confirming type-specific variation in the SU5 epitope. The number of potential N-linked glycosylation sites (PNGS) ranged from 3 to 6 in respective sequences and were located in different positions. The analysis of LTR sequences revealed that sequences corresponding to the TATA box, AP-4, AML-vis, and polyadenylation signal (poly A) were quite conserved, while considerable alteration was observed in AP-1 sites. Interestingly, our results revealed that all sequences belonging to subtype A17 had unique substitution T to A in the fifth position of TATA box and did not have a 11 nt deletion in the R region which was noted in other sequences from Poland. These data revealed a complex picture of SRLVs population with ovine and caprine strains belonging to group A and B. We present strong and multiple evidence of dually infected sheep and goats in mixed flocks and present evidence that these viruses can recombine in vivo.

## 1. Introduction

Small ruminant lentiviruses (SRLVs) include two retroviruses, Caprine arthritis encephalitis virus (CAEV), and Maedi-visna virus (MVV), members of the genus Lentivirus of the Retroviridae family. Originally, MVV and CAEV were considered as distinct viral species restricted to sheep and goats, respectively, but several reports indicated that there are different lentiviral subtypes able to infect both sheep and goats [1]. SRLVs induce a multisystem disease with progressive and debilitating inflammatory lesions in the mammary gland, joints, lungs, and the brain. Diseases caused by SRLVs may take severe clinical course; however, clinical signs develop after a several-year-long period and only in one third of the infected animals. On rare occasions, young children may develop a leukoencephalomyelitis with CNS signs [2]. Both asymptomatic and symptomatic animals can transmit the virus. Infection takes place through the ingestion of infected colostrum/milk and/or by direct contact through respiratory exudates from infected animals [3,4]. The incidence of SRLV infections causes economic losses and welfare problems in small ruminant production, since no therapy or vaccine is currently available.

The genome of SRLVs, which is integrated into host cells in the form of a provirus, is composes of three gene’s coding for structural proteins, *gag*, *pol*, and *env*, and additional open reading frames (*vpr-like*, *rev*, and *vif*), which encode for nonstructural proteins with regulatory functions in virus replication [5]. The provirus genome is flanked by repeated sequences known as long terminal repeats (LTRs) composed of three regions, U3, R, and U5. U3 region contains promoter sequence and different transcription factor binding sites, like AP-1, AP-4, AML (vis), and GAS, and play a regulatory role in transcription, integration, and polyadenylation of viral RNA [6,7]. The *gag* gene encodes the internal structural proteins, capsid (CA) protein (p25), the nucleocapsid (NC) protein (p14), and the matrix (MA) protein (p17). The capsid protein contains linear epitopes that induce antibody production and for this reason it is used for serological diagnostic tests. In addition, because it is most conserved and is also commonly used as the target fragment for phylogenetic analysis and genomic characteristics of SRLV strains. The *env* gene encodes two proteins inserted in an envelope, the surface (SU) protein (gp135) and the transmembrane TM protein (gp46). SU also stimulates the production of antibodies but is genetically variable and determines the antigenic variability of the strains [8].

SRLVs are characterized by a high degree of genetic variability leading to the variety of divergent strains and quasispecies. SRLV isolates have been classified into five genetic groups, A-E, and further divided into several subtypes. Group A is the most heterogenous group and has been subdivided into 22 subtypes (A1-A22), so far. Group B contains five subtypes (B1-B5) while group E contain to subtypes (E1 and E2) [9,10]. Group A and B refer to MVV-like and CAEV-like viruses, respectively. Viruses belonging to these two groups are the most predominant strains around the world while isolates belonging to groups C, D and E are geographically restricted to limited areas. Groups C and E refer to Norwegian and Italian strains, respectively, while group D was found in a few isolates from Switzerland and Spain, but they are now re-classified as group A [11,12,13,14,15].

Many studies have documented the circulation of different subtypes of SRLVs in both sheep and goats due to interspecies transmission. In fact, the majority of subtypes can cross the species barrier between sheep and goats under field conditions [1,16,17]. Early and accurate diagnosis of SRLVs is crucial for prevention of infection and disease control. However, antigenic variation of SRLVs poses a relevant problem for SRLVs testing and diagnosis since heterogeneity of circulating strains may be wider than those covered by available ELISA tests making serological response not always detectable. Therefore, genotype and subtype surveys of the circulating SRLV strains in each country should be constantly updated to ensure reliability of diagnostic tests. In particular, mixed flocks where sheep and goats live in close contact, represent a suitable environment to evaluate the degree of SRLVs heterogeneity due to the growing evidence of increased cross-species transmission in mixed flocks and possible recombination events [18,19,20].

In Poland, a seroprevalence study confirmed the presence of SRLVs in 33.3% and 71.9% of sheep and goat flocks, respectively [20,21,22]. SRLVs isolated so far from sheep and goats in Poland belonged to the well-known subtypes B1, B2, A1, A5, and A16, as well as subtypes A12, A13, A17, and A18 detected only in Poland [23,24,25,26]. In a previous study we showed that SRLV strains found in mixed flocks in Poland belonged to subtype A13 and A18, but this study was restricted to only the few flocks from the Małopolska region. In the present study, we aimed at extending the previous investigation carrying out genetic characterization and phylogenetic analysis of SRLVs from mixed flocks including a larger numbers of animals, and flocks from different parts of Poland. Genetic analysis of SRLVs from mixed flocks may help to understand the genetic and antigenic make-up of these viruses, phylogenetic relationship and their allocation into the recently established groups. Moreover, genetic studies may also be useful for the development of regionally-tailored diagnostic tests.

## 2. Materials and Methods

### 2.1. Animals and Samples

A total of 263 samples were investigated in this study, 163 from sheep and 100 from goats, and originating from 17 mixed flocks and different geographic regions of Poland. Sheep and goats were housed in the same barn with the possibility of direct contact and via water and feed troughs. Animals were randomly chosen from the flocks for serological study and were clinically healthy, without any clinical signs. Blood was taken in EDTA and serum tubes for serology and molecular analysis. Sera samples were tested for MVV/CAEV antibodies using the commercially test ID Screen MVV/CAEV Indirect Screening (IDvet, Grabels, France). EDTA-anticoagulated blood was used as a source of PBLs, which were isolated according to standard protocols [24]. The genomic DNA was extracted using a NucleoSpin Blood Quick Pure Kit (Macherey-Nagel GmbH & Co. KG, Dueren, Germany), according to the manufacturer’s recommendation. The quality and quantity of DNA was evaluated in a Nanophotometer (Implen GmbH, Munich, Germany). All methods were performed in accordance with the relevant guidelines and regulations. Specifically, blood collection was approved (no. 37/2016) by the Local Ethical Committee on Animal Testing at the University of Life Sciences in Lublin (Lublin, Poland).

### 2.2. PCR Technique

The CA (625 bp) fragment of the *gag* gene fragment, the V4V5 fragment of *env* gene, and the U3-R fragment of the LTR region were amplified by nested PCR. *Env* fragments were amplified with the combination of 423/564 primers, followed by nested PCRs using either 423/425, 563/564, 563/425, or 567/564. The PCR reactions were performed as previously described [26,27,28] (Table 1).

### 2.3. DNA Sequencing and Analysis

PCR products were purified using NucleoSpin Gel and PCR Clean-up (Marcherey-Nagel, GmbH 7 Co, Hamburg, Germany) and cloned into the pDRIVE vector (Qiagen, GmbH, Hilden, Germany). Ligation products were used to transform EZ Competent Cells (Qiagen, GmbH, Hilden, Germany) and plasmid DNA was extracted using the NucleoSpin Plasmid kit (Marcherey-Nagel, GmbH 7 Co., Hamburg, Germany). A minimum of five clones derived from each DNA sample were sequenced on a 3730 xl DNA Analyzer (Applied Biosystems, Foster City, CA, USA) using BigDye Terminator v3.1 Cycle Sequencing kit. The obtained SRLV sequences were trimmed and analyzed using the Geneious Pro 5.3 software (Biomatters Ltd., Auckland, New Zealand). All novel sequences reported in this study were submitted to the Gen-Bank database under accession numbers: OL348000- OL348058 for *gag* sequences and OL436259- OL436303 for env sequences. The evolutionary relationships of analyzed strains with other published sequences were investigated by constructing the phylogenetic trees from multiple alignments. The available sequences of the reference SRLV strains of genotypes A-C and E, represented isolates from a wide range of countries, were included in the analysis. In the present study, the SRLVs found by Colitti et al. [13] were renamed from A18 to A19 and from A19 to A20. All sequences were aligned using MUSCLE. Model testing was performed to select the best evolutionary model based on the Bayesian information criterion (BIC) and Akaike information criterion (AIC). According to the results General Time Reversible (GTR) model with the gamma distribution (+G) with 5 rate categories and by assuming that a certain fraction of sites are evolutionarily invariable (+I) was applied to infer a phylogenetic tree using maximum likelihood (ML) and neighbor-joining method. The reliability of the phylogenetic relationships was evaluated by nonparametric bootstrap analysis with 1000 iterations. Alignment, model testing, and tree building were performed using MEGA 6 application [29]. The tree topology was confirmed using the Bayesian method with the GTR model implemented in Genious software. Nucleotide and amino acid sequence percent identity (percentage of bases/residues which are identical) was estimated using Geneious software while pairwise genetic distances were calculated with the MEGA 6 software. The nonsynonymous (dn) and synonymous (ds) substitution rate was calculated using SNAP (Synonymous No-synonymous Analysis Program) v 2.1.1 [30]. Potential N-linked glycosylation sites were identified using the N-GlycoSite tool [31].

### 2.4. Analysis of Recombination

To detect possible recombination events, the Recombination Detection Program version 4 (RDP4) with default setting was used [32]. The software used seven primary exploratory recombination signal detection methods, RDP [33], GENECONV [34], BootScan [35], MaxChi [36], Chimaera [37], SiScan [38], and 3Seq [39]. The beginning and end breakpoints of the potential recombinant sequences were also defined by the RDP4 software. Putative recombinant events were considered significant when *p* ≤ 0.01 was observed for the same event using four or more algorithms.

## 3. Results

### 3.1. Phylogenetic Analysis

Out of 263 serum samples, 84 (32.0%) (53 from goats and 31 from sheep) originating from eight flocks were positive in the ELISA test. DNA extracted from the blood of serologically positive animals was used to amplify the CA fragment of the *gag* gene for phylogenetic analysis. Proviral DNA of 54 samples originated from 26 sheep and 28 goats from six different flocks was successfully amplified and sequenced (Table 2).

Obtained sequences were aligned with reference sequences representing the genotypes of SRLVs described to data, however, we included only sequences of appropriate length matching to data obtained in this study. An unrooted phylogenetic tree is shown in Figure 1.

Sequences of Polish strains analyzed in this study were widely distributed on the tree, clustering in subtype B2, A5, A12, and A17. In particular, sequences of sample s#21, s#20, g#9510, g#3540, s#14, g#0599, g#3535, g#0788, g#0580, and s#29 from flock 16 clustered within subtype B2 were closely related with Polish strains #11, #2437 and #4106, which were previously detected in the same Polish region (mean nt distance 2.6% ± 0.5%). Sequences originated from sheep #2590, #3691, #3275, #4315 from flock 13, from sheep #9855 from flock 14 and from sheep #0334 from flock 10 also clustered within subtype B2 but were more closely related with sequences of Spanish strain #496 and Swiss strain #5720 (mean nt distance 7.3% ± 1.8%). Sequences originating from goat #8039, #8046, #9692, #1318, and #8008 from flock 13 were closely related with Polish strains #6038, #5819, and #5826 (mean nt distance 1.3% ± 0.4%), representing subtype A5, while sequences originated from goat #9431, #8172, #6909, #5621, #1580, #1485, #5654, and #5686 from flock 17 were more closely related with Polish strains #5616, #8344, and #3085 representing subtype A17 (mean nt distance 2.6% ± 0.6%). Twenty sequences had been assigned to the A12 subtype and phylogenetic analysis clearly showed the existence of three separated subgroups within this cluster. Sequences from sheep #40, #12, #33, #3, #16, #1, #13, #6, #4 and #14 from flock 16 (I cluster) were closely related with sequences of Polish strains #4007, #4819 and #1202 with mean nt distance of 4.0% ± 1.4%. Sequences from goat #7219, #8891, #7102, #7134, #7096 and #6808 from flock 10 (II cluster) and sequences from goat #8699, #3533, #9509 and #3535, from flock 16 (III cluster) were closely related with sequences of Polish strains #15, #10 and #13 but they formed separated clusters (mean nt distance 7.9% ± 1.2%). Additionally, sequences from sheep #1622, #4315, #4018, #2590, and goat #8046, from flock 13 as well as sequences from sheep #5023, #3249, #3188, #3225, and #3201 formed new clusters within group A, which could be tentatively named as A23 and A24, respectively. Affiliation of these new clusters were supported with high bootstrap values ≥84. Sequences of the proposed subtype A23 and A24 had a mean sequence similarity (intra-subtype similarity) of 4.3% and 3.4%, respectively. The mean genetic distances between new subtypes and other subtypes representative for genotype A varied from 13.1% to 24.6% for subtype A23, and from 12.5% to 22.3% for subtype A24 (Table 3). To evaluate the robustness of our analysis, we also performed phylogenetic analysis using neighbor-joining and the Bayesian inference method (Appendix A), which resulted the same classification of all strains, whereby supporting the existence of new subtypes A23 and A24. Subtypes A5 and A17 have been found only in goats while subtype A24 has only been detected in sheep. In contrast, subtypes A12, A23 and B2 have been found in both sheep and goats. Dual infection with B2 and A12 was found in sheep #14 and goat #3535 from flock 16. Co-infection with B2/A23 was detected in sheep #2590 and sheep #4315 from flock 13 while co-infection with subtypes A5/A23 was detected in goat #8046, also originating from flock 13. Only in two flocks detected SRLV sequences representing one genotype. In flock 12 circulated only subtype A24 while in flock 17, subtype A17. This was confirmed by pairwise nucleotide comparison, in which distances estimated among sequences derived from flocks 12 and 17 varied from 1% to 2.9% and from 0% to 4.3%, respectively. On the other hand, in four flocks highly divergent SRLV subtypes were found: subtypes B2/A12 in flock 10, subtypes A5/A23/B2 in flock 13, subtypes A24/B2 in flock 14, and subtypes B2/A12 in flock 16.

All samples were also used to amplify 608 bp fragment of *env* gene. Out of 54 tested samples, 45 (20 from sheep and 25 from goats) were successfully amplified, and after sequencing were subjected to phylogenetic analysis. The phylogenetic tree (Figure 2) confirmed that Polish sequences belonged to subtype B2, A5, A12, and A17, as well as to new identified subtypes A23 and A24. Similarly to the *gag* phylogenetic assignment, *env* sequences from sheep #2590, #4315, #3275, and #1622 from flock 13 formed the new subtype A23, while sequences from sheep #3188, #3249 and #3201 from flock 12 formed subtype A24. The mean nucleotide distance between sequences belonging to the A23 subtype and those belonging to A24 subtype was 22.0%, while the mean distance between these subtypes and other subtypes within group A ranged from 18.5% to 27.8% and from 20.4% to 27.6% for A23 and A24, respectively. Sequences from sheep #9855, #3691 and #4018 previously located in subtypes B2 and A23, respectively, now created a separate branch clustered closely with North American MVV strains 85/34 and S93. In addition, sequences from sheep #5023, which on the basis of *gag* fragment was affiliated to subtype A23, now formed a separate cluster.

### 3.2. Identification of Putative Recombination

A recombination analysis of sequences used for phylogenetic analysis was performed to verify if sequences obtained in this study resulted from a recombination of already known sequences. On the basis of *gag* alignment, one putative recombination event was detected by five statistical methods with high significance and reliability. The recombinant sequence #14(2) was detected in sheep co-infected with strains A12/B2 from flock 16. In this recombination event, the beginning and ending breakpoints were located at 27 and 320 nucleotides in alignments and the major and minor parents were #4007, representing subtype A12 (90.9% similarity) and #0599 representing subtype B2 (100% similarity), respectively (Figure 3a). On the basis of *env* alignment, two putative recombination events were detected. Four methods detected a recombination event in #13s4018 between positions 64 and 276 in alignment with #13s4315 (subtype A23) as the minor parent and unknown, suggesting #5819 (subtype A5), as the major parent. The results also indicated that #13s3691 arose from recombination events between the same breakpoints position, 64 and 276 in alignment, but with #13s4315 (subtype A23) and unknown, (suggesting #5819 A5), as the major and minor parents, respectively (Figure 3b).

### 3.3. Analysis of Immunodominant Regions

To analyze the conservation of immunodominant regions of sequences of Polish strains analyzed in this study, their deduced amino acid (aa) sequences of capsid and surface proteins were aligned with the aa sequences of reference parental strains, Cork and K1514. Pairwise percent identity of the *gag* amino acid sequences of Polish SRLVs was high and ranged from 81.3% to 100%. Furthermore, Polish sequences shared 82.1–96.3% and 92.4–97.7% amino acid sequence identity with strains Cork and K1514, respectively. Analysis revealed that all *gag* sequences belonging to group B had glycine-glycine (GG) motifs and all MVV-like sequences had asparagine-valine (NV) motif, typical for strains belonging to group A (Appendix A). Immunodominant epitope 3, situated at the C-terminal end of the capsid protein, was highly conserved between sequences belonging to group A and B, while the analysis of sequences in epitope 2 revealed a perfectly conserved region (GKLNEEAERW) located at the N-terminal part of epitope and distinct region at the C-part of the epitope specific for MVV-like (VRQNPPGP) and CAEV-like (RRNNPPPP) strains. In the Major Homology Region (MHR), which is usually highly conserved in the *gag* gene of all retroviruses, some alterations were present within group A (Figure 4). In particular, sequence from goat #9692 had isoleucine (I) instead of valine (V) at the fourth position compared to strain #K1514, while sequences from goat #6808 and #8699 had threonine (T) instead of asparagine (N), and valine (V) instead of isoleucine (I) at the eighth and 16th positions, respectively. All sequences representing subtype A17 and sequence from goat #6808 representing subtype A12 had substitutions arginine (R) instead of lysine (K) at the fifth position, while sequences from goat #3535 and #9509, representing subtype A12, had substitution arginine (R) instead of lysine (K) in the seventh position of MHR. All goat-derived sequences representing subtype A5 and sequences from goats #3535 and #9509 had substitution glutamic acid (E) instead of aspartic acid (D) at the 14th position. Six sequences had substitution serine (S) instead of threonine (T) and 35 sequences had substitution asparagine (N) instead of threonine (T) at the ninth position compared to the sequences of strain #K1514. Type B sequences had more conservative MHR sequences. Compared to sequences of strain Cork, all sequences had substitution at the eighth position (T/S or T/G). Sequences from sheep #0334 had substitution serine (S) instead of asparagine (N) and serine (S) instead of proline (P) at the ninth and eleventh positions. Five sequences representing subtype B2 which formed subclusters on the phylogenetic tree had unique substitution threonine (T) instead of alanine (A) at the 16th position of MHR.

The *env* aa sequences of Polish strains were more heterogenous than *gag* sequences showing 54.1–100% similarity to each other and 57.1–73.1% and 59.4–73.7% to the strains K1514 and Cork, respectively. Sequences of the variable region (V4) of analyzed strains differed significantly. Comparison of aa sequences in epitope SU5 revealed the conserved region (VRAYTYGV) located at the N-terminal part of the epitope. The variable region was conserved among the sequences of strains belonging to subtypes A5, A23, A24, and B2 showing type-specific variation. Sequences belonging to subtypes A12 and A17 showed intra-subtype variability (Figure 5). Sequences could be divided into groups corresponding groups formed on the phylogenetic tree.

To determine if the nucleotide sequences encoding Gag and Env proteins evolved under positive selective pressure, the ratio of nonsynonymous to synonymous base substitutions were estimated. The results showed that for both fragments, the dN/dS ratio for caprine and ovine sequences was below 1, showing a negative selection. Additionally, the number of potential N-linked glycosylation sites (PNGS) was estimated and ranged from 3 to 6 in respective sequences. In all caprine sequences of subtype A17 from flock 17, and in all caprine sequences of subtype A12 from flock 10, four potential N-linked glycosylation sites were detected, in positions 8, 14, 32, and 39 in the alignment. These four PNGS were also detected in ovine sequence #5023 from flock 14, which formed a new cluster within group A on the phylogenetic tree. In two out of three ovine sequences, representing subtype A24 from flock 14, and in three sequences (14s9855, 13s4018, and 13s3691) which formed a cluster distinct from known A and B subtypes additional glycosylation site, at position 53, was detected. In sequences from flocks 13 and 16, 3-6 N-linked glycosylation sites were observed, but with some heterogeneity in the positions between different isolates. In most ovine sequences representing subtype A23 from flock 13, four PNGS were detected (at positions 8, 14, 32, and 39), while in caprine sequences representing subtype A5, the number of PNGS ranged from 3 to 5 and were located at different positions. In most of the A12 sequences originating from the sheep from flock 16, four PNGS were detected (at positions 14, 32, 39, and 43), while in caprine sequences representing subtype A12, five PNGS were detected (at positions 8, 14, 32, 39, 43, and 52). B2 sequences derived from this flock had four PNGS at positions 8, 14, 32, and 39.

### 3.4. Analysis of LTR Sequences

LTR sequences from 47 out of 54 samples (87%), which were successfully amplified, were aligned with sequences of prototype strains K1514 and CAEV-Cork representative for SRLVs groups A and B, respectively. The nucleotide pairwise percent identity of Polish sequences ranged from 72.3% to 100%. Furthermore, Polish sequences shared 48.4–53.8% and 74.0–85.8% sequence identity with strains Cork and K1514, respectively. The LTR regions analyzed in this study contained two AP-1, one AML (vis) and one AP-4 putative motifs without any duplication or deletion in their U3 regions compared with reference sequences (Figure 6). Sequences corresponding to the TATA box, AP-4, and polyadenylation signal (poly A) were quite conserved. All sequences belonging to subtype A17 had unique substitution T to A in the fifth position of TATA box. Sequence of AML(vis) motif was present in all Polish samples and was identical with the sequence of K1514 strain, except for sequences originating from goats #8891 and #7219 which had substitution A to G in the second position of AML-vis. The AP-1 sites were less conserved. Sequences of the first AP-1 site were identical in all Polish sequences analyzed and the sequence of K1514 strain (TCATGTA), but differed from the sequence of Cork strain (TGACATA), while in the second AP-1 site considerable nucleotide changes were observed. All Polish sequences analyzed, except sequences representing subtype A17, had the specific 11 nt deletion in the R region. However, A17 sequences had the CCGAAGGAAAG insertion almost identical, like in the K1514 strain. Furthermore, all Polish sequences had 13 nt deletion in the U5 region.

## 4. Discussion

Previous studies revealed that the SRLVs population in Poland is highly heterogeneous. SRLVs isolated so far from sheep and goats in Poland belonged to the well-known subtypes B1, B2, A1, A5, and A16, as well as subtypes A12, A13, A17, and A18—detected only in Poland. Since mixed flocks promote interspecies transmission and the emergence of new variants [18,19,20], the aim of this study was to perform genetic characterization of field SRLV strains present in Polish mixed flocks to get a better insight into their heterogeneity.

The current SRLVs phylogeny consisting of five main groups, which are divided in multiple subtypes, emphasizes the high genetic diversity among SRLV strains. In 2004, Shah et al. proposed a classification of SRLVs based on sequences of the *gag-pol* (1.8 kb) [40]. However, due to low proviral load and high genetic variability of SRLV strains, in many cases this fragment could not be obtained [12,18,41,42]. As a result, classification of SRLVs is more often performed on a very conservative ~0.4 kb *gag* fragment for which sequences representing most of subtypes are available. Using this fragment, we confirmed circulation of subtypes B2, A5, A12, and A17 in analyzed flocks and revealed circulation of new subtypes A23 and A24. These new subtypes were closely related to subtypes A13 and A18 which were previously detected only in Poland, suggesting their common origin. Clearly separation of these subtypes from other Polish subtypes, located in distinct clusters on the phylogenetic tree, suggests the presence of at least three SRLV genetic lines circulating in Poland which are independently evolving. As was previously reported, some strains can cluster to different subtypes depending on the fragment that is analyzed. Such an observation was noted for the subtype A19, A20, and B5 strains ([10], this report). This suggests that these strains could be generated by recombination. Thus, to confirm the existence of our putative new subtypes, we decided to perform phylogenetic analysis using variable *env* sequences, which confirmed that they formed a separate clusters with no clear relation to any of the previously described group A subtypes.

Lentiviral genomes are among the most rapidly evolving known. Most of the mutations are introduced during the reverse transcription stage of the viral life cycle as a consequence of low fidelity of reverse transcriptase, which has no proofreading activity. However, interspecies transmission, co-infections, and recombinations are the main mechanisms which contribute significantly to genetic variability and accelerate viral evolution [17,43,44,45]. Direct evidence of interspecies transmission of SRLVs from sheep to goats and vice versa has been documented by the detection of most subtypes in both sheep and goats [16,18,24,46]. It is also evidenced that mixed flocks, where sheep and goats are kept together, is the factor promoting cross-species infections which can result in the emergence of new variants [18,19,20]. In our study, subtypes A5 and A17 have been found only in goats, subtype A24 has been detected only in sheep, while subtypes A12, A23, and B2 have been found in both sheep and goats. This confirms the ability of SRLVs to frequently cross the species barrier under natural conditions. Furthermore, co-infection with strains belonging to different subtypes was evidenced in three sheep and two goats which originated from two flocks. Sheep #14 and goat #3535 from flock 16 were co-infected with B2/A12, while two sheep (#2590 and #4315) and one goat (#8046) originated from flock 13 were co-infected with B2/A23 and A5/A23, respectively. Existence of co-infected animals can be explained by circulation of more than one subtype in these flocks. Our results revealed that in four out of six flocks, highly divergent SRLV subtypes were found (subtypes B2/A12, A5/A23/B2, A24/B2, and B2/A12). In flock 13, circulation of sequences belonging to subtypes A5, A23 and B2 were found. All goats were infected with strains representing subtype A5, while in sheep, only sequences belonging to subtypes A23 and B2 were found. Only one goat was co-infected with A5/A23 which strongly indicates that the direction of transmission of subtype A23 in this case was from sheep to goat. Mean *gag* nucleotide distance of A23 sequences found in this co-infected goat and A23 sequences found in sheep was 3.5%—strongly confirming their common origin. Furthermore, two sheep from this flock were co-infected with subtypes A23 and B2. However, it is difficult to explain which subtype was first introduced to the flock, because both subtypes were detected at the same frequency. The *gag* mean nucleotide distance of B2 and A23 sequences was 0.5% and 3.9%, respectively, while the mean nucleotide distance between these subtypes was 32.9%. This confirms the circulation of two distinct subtypes in the sheep from this flock which do not represent the evolution of the homologous strains. In flock 16, circulation of two subtypes, A12 and B2, was detected in both, sheep and goats. Because the most of sheep were infected with subtype A12 and most of goats were infected with subtype B2, the transmission of these subtypes is suggested to be from goats to sheep and from sheep to goats, respectively. Interspecies transmission was also observed in flock 10 and flock 14, where subtypes A12/B2 and B3/A24 were found. The high genetic distance of sequences found in these flocks strongly suggests that they originated from different sources. The introduction of different subtypes of SRLVs in analyzed flocks could not be tracked back because we don’t know the history of animal movements, but they most likely resulted from the purchase of infected animals from other flocks. The introduction of new animals to a flock for genetic improvement is common practice in Poland and undoubtedly represents a high risk factor for the spread of SRLVs, which may result in higher diversity of SRLVs [40,47,48]. In Poland, it is also facilitated by the lack of SRLVs eradication programs and any veterinary controls.

Co-infection with at least two subtypes offers opportunities for viral recombination which is believed to be a powerful source of genetic variability of lentiviruses leading to the emergence of new strains. Previous studies have provided clear evidence of recombination between SRLVs belonging to groups A and B, as well as between subtypes belonging to the same group [18,20,24,45,46,49,50]. In the present study, three examples of recombinations have been found. One putative recombination event was detected in the *gag* fragment in sheep #14 co-infected with strains A12/B2, while on the basis of *env* fragments, two putative recombination events were detected in two sheep from flock 13. Recombination is a frequent event in the envelope gene which could lead to the generation of chimerical SRLVs with altered cell tropism, pathogenicity and transmission efficiency which may endanger not only domestic ruminants but also other animal species [51,52,53].

Analysis of sequences is important, not only for evaluating the spread of SRLV subtypes but also for gaining knowledge of antigenic variability. Alterations in the amino acid sequences of immunodominant epitopes determine their antigenicity and may impact on sensitivity of serological tests. The primers used in this study allowed the sequencing of two immunodominant epitopes of capsid protein—epitopes 2 and 3, which were identified by Rosati et al. [54] and which are used in SRLVs ELISA tests. The CA is the major viral core protein and antibodies against this antigen are usually first generated in sheep and goats infected with SRLVs and remain detectable for a long time [55]. Our results, which are consistent with other reports, showed conservation of epitope 3 and the amino-terminal part of epitope 2 (GKLNEEAERW), as well as the group-specific part located at the C-termini of epitope 2 (group A, VRQNPPGP, group B, RRNNPPPP) [24,25,56,57]. More variability was found in the MHR, which is usually highly conserved in many retroviruses [12,58,59]. MHR of Polish MVV-like sequences had more alteration and revealed unique mutations in sequences of subtypes A5 and A17. All Polish sequences representing subtype A17 had unique substitution lysine (K) to arginine (R) at the fifth position of MHR, while substitution aspartic acid (D) to glutamic acid (E) at the 14th position of MHR was exclusively found in Polish sequences representing subtype A5. Because these subtypes were found in goats, we hypothesize that these changes may have resulted from cross-species transmission, as many genetic changes after SRLVs cross-species transmission were reported in many publications [1,16,20,40,51]. However, subtypes A17 and A5 were detected only in goats of Polish White Improved/Polish Fawn Improved and Carpathian breeds, respectively. This may also suggest that mentioned changes could have arisen as a result of long host-virus adaptation and evolution.

As expected, the SU sequences showed more extensive variations in comparison to CA sequences. Previous sequence analysis of SRLV strains defined five major variable regions of SU [60], but sequences obtained in this study allowed for the analysis of only the V4 and V5 regions. Although the dN/dS ratio for V4V5 sequences was below 1, indicating the existence of a purifying selection, we noticed that some regions may be under positive selection. Our results revealed that the V4 region of Polish strains differed significantly, and that the differences mainly occurred within a region previously proposed to be part of a variable, conformational neutralization epitope [61,62]. Moreover, insertion and deletion only occurred in highly variable region (HV2) [61], confirming that this region underwent rapid sequence evolution during SRLVs infection. Mutation in this region resulted in escape from neutralization and the creation of a new type of neutralization specificity [62]. Thus, it is suggested that the V4 region may have an analogous function to V3 in HIV-1 since the V3 loop of HIV-1 is the major target for neutralizing antibodies [45,63]. Interestingly, in the V4 region, a ’signature pattern’ related to different clinical status in sheep and goats has been found [64]. Comparison of aa sequences in epitope SU5 revealed conserved region (VRAYTYGV) located at the N-terminal part of the epitope and highly variable motif at the C- terminal which was well-conserved among strains belonging to the same subtypes confirming that the SU5 epitope is responsible for type-specific immune response allowing strain-specific diagnosis. Moreover, this observation appears to support the hypothesis that this region may function as a decoy antigen and, therefore, in a given subtype, evolves under negative selection [65]. Additionally, the V4V5 regions contain the majority of the conserved N-linked glycosylation sites and cysteine residues, suggesting that they form a highly constrained and surface-exposed domain [60]. These sugars, or “glycans,” play several important roles in the infective cycle of the viruses. In HIV-1, effects of glycosylation on viral replication, glycoprotein cleavage, CD4 binding activity, and coreceptor usage have been documented. Removal of the glycan may indirectly increase viral activity by causing a shift in the V3 loop, leading to an increase in co-receptor binding while high density of glycans protect the virus against neutralizing antibodies as a “glycan shield” [66,67]. In this study, we observed differences in the number of potential N-linked glycosylation sites (PNGS) which ranged from 3 to 6 in respective sequences. Most of the sequences had four PNGS which were detected, at positions 8, 14, 32, and 39 in the alignment, suggesting that these PNGSs may be evolutionarily conserved. Although some glycans are evolutionarily conserved, the number of others may vary extensively within infected individuals, since they may appear or disappear over the course of an infection in a single host [66,67]. Thus, variation in the number and position of PNGSs in Polish sequences may result from a host-species adaptation of SRLVs, as was evidenced for HIV-1. In general, the average numbers of predicted glycosylated positions were relatively conserved between the different subtypes of SRLVs.

U3-R regions contain elements described as important for the regulation of SRLVs transcription and replication. Therefore, sequence variation in LTRs might affect the interactions with cellular factors and alter viral expression and replication. Furthermore, it was also demonstrated that LTR sequence variability may affect tissue tropism and disease outcome [63]. The analysis of Polish LTR sequences revealed that sequences corresponding to the TATA box, AP-4, AML-vis, and polyadenylation signal (poly A) were quite conserved which was previously reported in strains from different geographic areas. This sequence conservation argues in favor of their importance in the replication strategies of SRLVs [12,26,42,68,69,70]. On the other hand, considerable alteration was observed in AP-1 sites, which confirmed previous findings suggesting that AP-1 sites may by functional even despite some changes [70,71]. AP-1 binding sites are important for regulation of SRLVs expression in macrophages and are required for phorbol ester-inducible gene expression of SRLV. Multiple copies of AP_1 sites presumably allow transcription regardless of mutations [69,70,71]. The TATA box is present and highly conserved in all retroviruses [70]. What is interesting is that our results revealed that all sequences belonging to subtype A17 had unique substitution T to A in the fifth position of the TATA box. Mutation in the TATA box of HIV-1 sequence changes the binding of the TATA-binding protein (TBP) which leads to a decrease in transcription [72,73]. The meaning of mutation detected in TATA box of subtype A17 SRLVs is unknown and warrants further study. Additionally, our results revealed that all sequences representing subtype A17 did not have a 11 nt deletion in the R region, which was noted for other sequences from Poland. Correlation between this deletion and the occurrence of clinical signs in infected animals has been suggested [51,69,74], but our results did not confirm these assumptions. All animals infected with both virus carrying the deletion and without deletions, were without clinical signs.

In conclusion, the results of this work extend the current knowledge on the distribution of SRLV subtypes in sheep and goats from Poland. Our results showed that SRLVs circulating in Poland are highly heterogeneous with ovine and caprine strains belonging to group A and B. We present strong and multiple evidence of dually infected sheep and goats in mixed flocks, and present evidence that these viruses can recombine in vivo. The results of the phylogenetic analysis revealed the existence of putative new subtypes which should lead to the consideration of an update of current SRLVs classification. Furthermore, genetic analysis of Polish SRLV sequences revealed some specific alteration present in *gag* and LTR gene fragments in some subtypes. Thus, the isolation and characterization of biological properties of these viruses should be performed to evaluate their pathogenic potential.

## Figures and Tables

**Figure 1 viruses-13-02529-f001:**
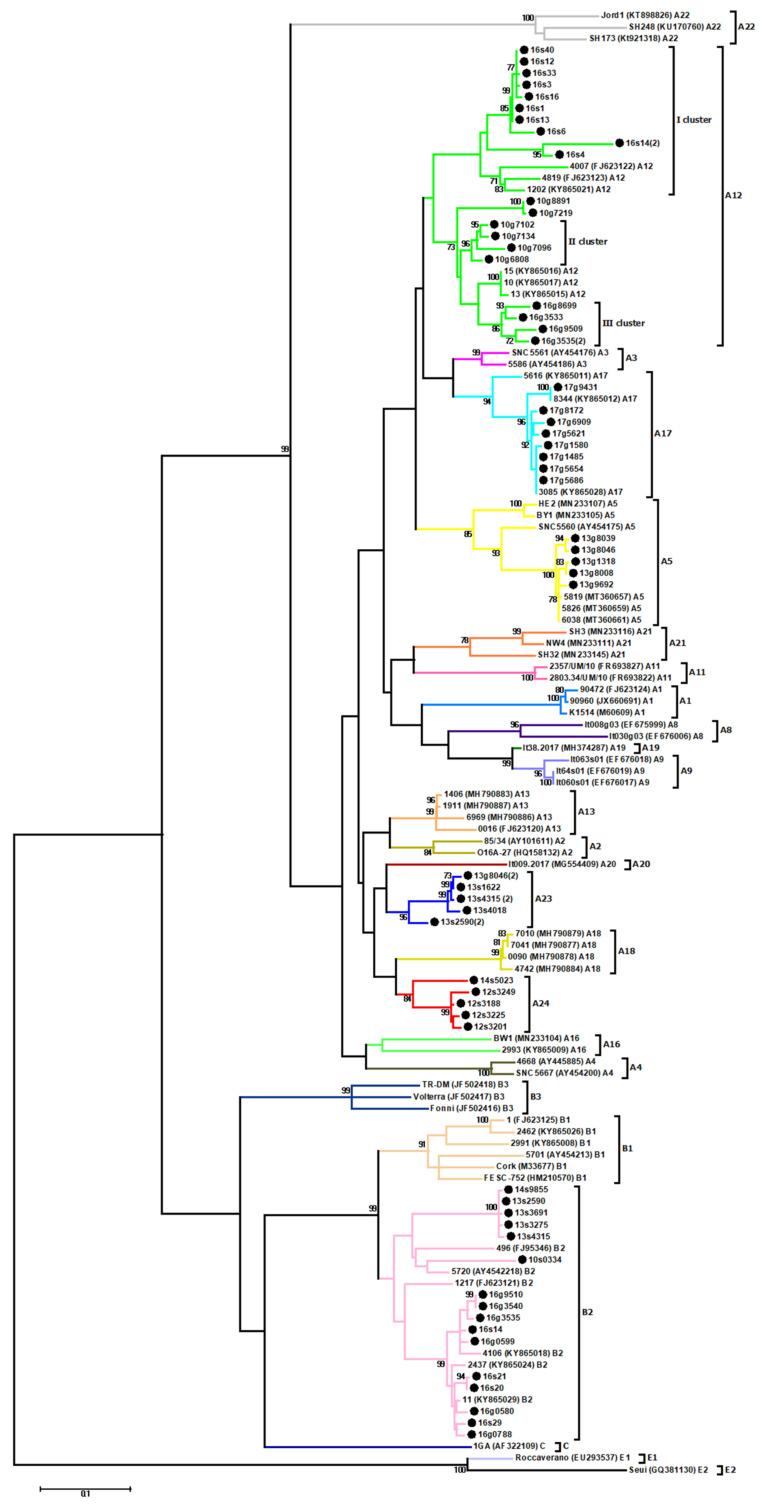
Maximum-likelihood phylogenetic tree based on alignment of CA fragment of gag gene. Sequences from this study are labeled by black circles and their names are proceeded by the flock origin and the animal species (s-sheep; g-goat). Subtypes are marked in different colors. Numbers at the branches indicate the percentage of bootstrap values obtained from 1000 replicates. Bootstrap values >70% are shown.

**Figure 2 viruses-13-02529-f002:**
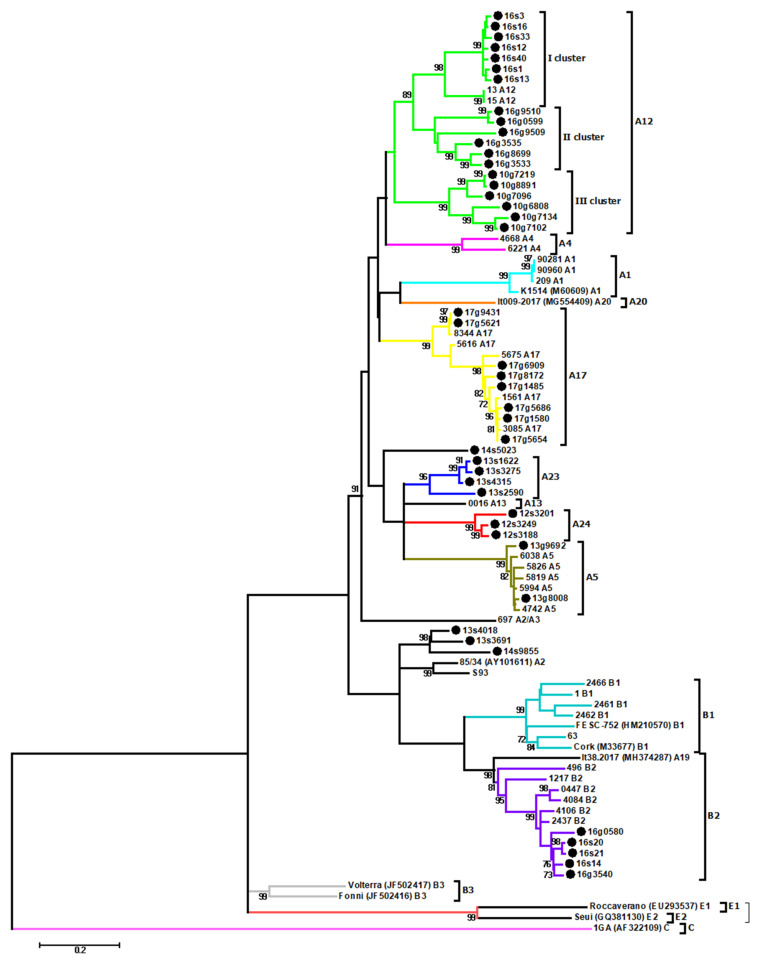
Maximum-likelihood phylogenetic tree based on alignment of the fragment of *env* gene. Sequences from this study are labeled by black circles and their names are proceeded by the flock origin and the animal species (s-sheep; g-goat). Subtypes are marked using different colors. Numbers at the branches indicate the percentage of bootstrap values obtained from 1000 replicates. Bootstrap values >70% are shown.

**Figure 3 viruses-13-02529-f003:**
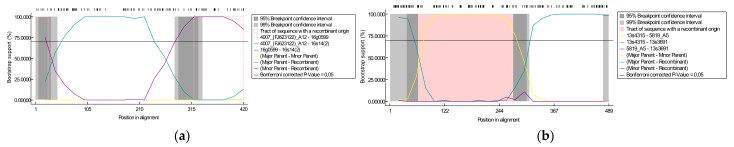
The BootScan analysis of recombination in the *gag* (**a**) and *env* (**b**) alignments. The analysis was performed with the pairwise distance model with a window size of 200, step size of 20, and 1000 bootstrap replicates by the RPD4 program.

**Figure 4 viruses-13-02529-f004:**
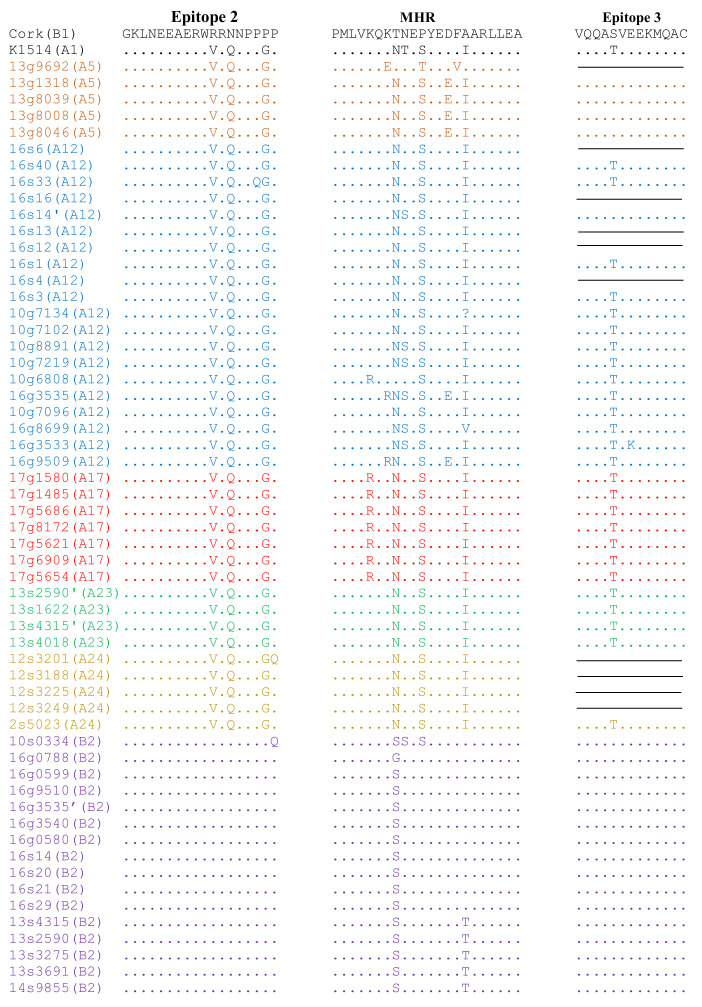
Alignment of deduced amino acid sequences of immunodominant epitopes of capsid protein and major homology region (MHR) of the SRLVs obtained in this study and K1514 (GenBank accession number M60609) and Cork (GenBank accession number M33677) reference strains, which are MVV (group A) and CAEV (group B) prototype strains, respectively. Identical residues are indicated by dots (.).

**Figure 5 viruses-13-02529-f005:**
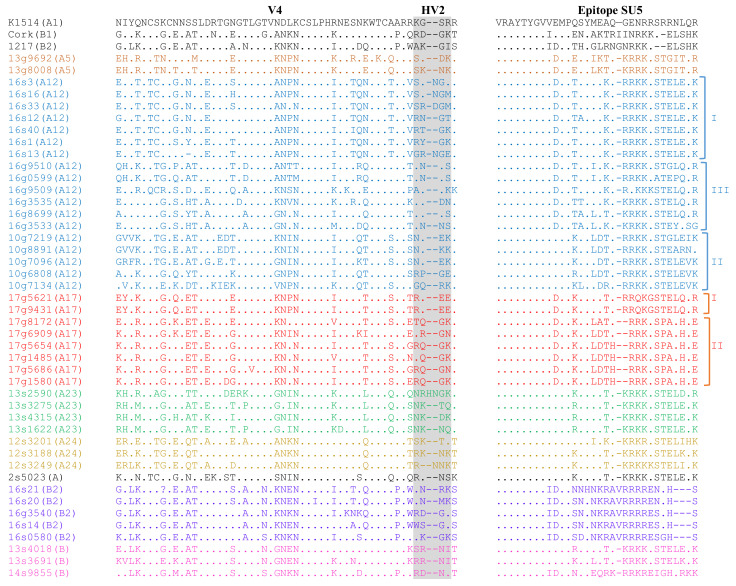
Alignment of deduced amino acid sequences of immunodominant epitope of ENV protein and variable region V4 of the SRLVs obtained in this study and K1514 (GenBank accession number M60609) and Cork (GenBank accession number M33677) reference strains, which are MVV (group A) and CAEV (group B) prototype strains, respectively. Deletions are indicated by a dash (-) and identical residues are indicated by dots (.).

**Figure 6 viruses-13-02529-f006:**
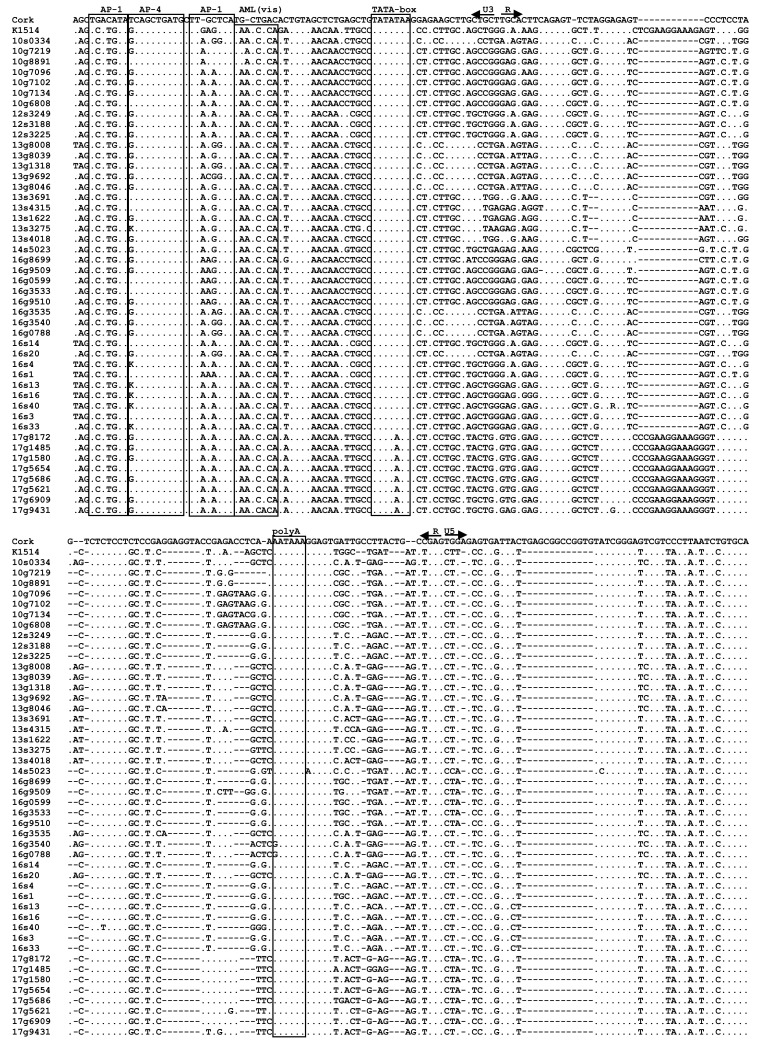
Alignment of nucleotide sequences of the LTR region of the Polish SRLV strains. Sequences are aligned against prototype strains K1514 (GenBank accession number M60609) and Cork (GenBank accession number M33677) representative for SRLV group A and B, respectively. Dots indicate identity with Cork and dashes represent gaps. The boundaries between U3, R, and U5 are indicated by straight arrows. AP-1, AP-4, AML (vis) motifs, the TATA box, and polyadenylation signal (poly A) are marked by boxes.

**Table 1 viruses-13-02529-t001:** Primers pair used for PCRs.

Pair	Primers	Sequences (5′-3′)	Orientation	PCR	Product Length (bp)
	LTR				
A	LTREFW	ACTGTCAGGRCAGAGAACARATGCC	F	1	407
	LTRERV	CTCTCTTACCTTACTTCAGG	R	1	
B	LTRIFW	AAGTCATGTAKCAGCTGATGCTT	F	2	213
	LTRIRV	TTGCACGGAATTAGTAACG	R	2	
	*Env*				
C	423	GGRGCAGARATMATHCCWGAARVYHTGA	F	1	1205
	564	GCYAYATGCTGIACCATGGCATA	R	1	
D	423	GGRGCAGARATMATHCCWGAARVYHTGA	F	2	1076
	425	CCTGCRGCAGCYAYTATHGCCAT	R	2	
E	563	GAYATGRYRGARCAYATGAC	F	2	818
	564	GCYAYATGCTGIACCATGGCATA	R	2	
F	563	GAYATGRYRGARCAYATGAC	F	2	689
	425	CCTGCRGCAGCYAYTATHGCCAT	R	2	
G	567	GGIACIAAIACWAATTGGAC	F	2	608
	564	GCYAYATGCTGIACCATGGCATA	R	2	
	*Gag*				
H	GAGf1	TGGTGARKCTAGMTAGAGACATGG	F	1	1350
	P15	GTTATTCCATAGGAGGAGCGGACGGCACCA	R	1	
I	CAGAG5	GCRGGRGGGAAGRAGYTGGAA	F	2	625
	CAGAG3	ACATGCTTGCATTTTTTYTTCTAC	R	2	

**Table 2 viruses-13-02529-t002:** Information on SRLV sequences obtained from sheep/goats originated from Polish mixed flocks.

					GAG		ENV	
Sample No.	Flock	Region	Host	Strain	GenBank Accession Number	Proposed Subtype	GenBank Accession Number	Proposed Subtype
1	10	Wielkopolskie	goat	7134	OL348032	A12	OL436271	A12
2			goat	7102	OL348031	A12	OL436303	A12
3			goat	6808	OL348029	A12	OL436270	A12
4			goat	7219	OL348023	A12	OL436269	A12
5			goat	7096	OL348030	A12	OL436268	A12
6			goat	8891	OL348024	A12	OL436297	A12
7			sheep	0334	OL348005	B2	N/A	N/A
8	12	Podkarpackie	sheep	3225	OL348051	A24	N/A	N/A
9			sheep	3201	OL348052	A24	OL436300	A24
10			sheep	3188	OL348050	A24	OL436296	A24
11			sheep	3249	OL348049	A24	OL436302	A24
12	13	Podkarpackie	goat	1318	OL348020	A5	N/A	N/A
13			goat	8008	OL348021	A5	OL436267	A5
14			goat	9692	OL348019	A5	OL436266	A5
15			goat	8039	OL348017	A5	N/A	N/A
16			goat	8046	OL348018, OL348056	A5/A23	N/A	N/A
17			sheep	4315	OL348001, OL348058	A23/B2	OL436285	A23
18			sheep	1622	OL348057	A23	OL436287	A23
19			sheep	4018	OL348055	A23	OL436260	B
20			sheep	2590	OL348004, OL348054	A23/B2	OL436284	A23
21			sheep	3691	OL348002	B2	OL436261	B
22			sheep	3275	OL348003	B2	OL436286	A23
23	14	Podkarpackie	sheep	9855	OL348000	B2	OL436298	B
24			sheep	5023	OL348053	A24	OL436272	A
25	16	Lubelskie	sheep	40	OL348037	A12	OL436276	A12
26			sheep	33	OL348035	A12	OL436299	A12
27			sheep	3	OL348036	A12	OL436259	A12
28			sheep	12	OL348038	A12	OL436275	A12
29			sheep	16	OL348039	A12	OL436277	A12
30			sheep	1	OL348034	A12	OL436273	A12
31			sheep	13	OL348040	A12	OL436274	A12
32			sheep	6	OL348033	A12	N/A	N/A
33			sheep	4	OL348022	A12	N/A	N/A
34			sheep	14	OL348006, OL348016	A12/B2	OL436264	B2
35			sheep	21	OL348014	B2	OL436301	B2
36			sheep	20	OL348015	B2	OL436263	B2
37			sheep	29	OL348011	B2	N/A	N/A
38			goat	8699	OL348025	A12	OL436282	A12
39			goat	3533	OL348026	A12	OL436283	A12
40			goat	3535	OL348010, OL348028	A12/B2	OL436281	A12
41			goat	9509	OL348027	A12	OL436280	A12
42			goat	9510	OL348008	B2	OL436278	A12
43			goat	3540	OL348009	B2	OL436265	B2
44			goat	0599	OL348007	B2	OL436279	A
45			goat	0788	OL348012	B2	N/A	N/A
46			goat	0580	OL348013	B2	OL436262	B2
47	17	Mazowieckie	goat	1485	OL348044	A17	OL436290	A17
48			goat	5654	OL348045	A17	OL436292	A17
49			goat	5686	OL348046	A17	OL436291	A17
50			goat	1580	OL348043	A17	OL436293	A17
51			goat	5621	OL348048	A17	OL436295	A17
52			goat	6909	OL348047	A17	OL436288	A17
53			goat	8172	OL348042	A17	OL436289	A17
54			goat	9431	OL348041	A17	OL436294	A17

N/A—not available.

**Table 3 viruses-13-02529-t003:** Estimated of mean evolutionary divergence between subtypes of genotype A (inter-genotype) based on the CA fragment of *gag* gene.

	A1	A2	A3	A4	A5	A8	A9	A11	A12	A13	A16	A17	A18	A19	A20	A21	A22	A23	A24
A1	-	-	-	-	-	-	-	-	-	-	-	-	-	-	-	-	-	-	-
A2	18.6	-	-	-	-	-	-	-	-	-	-	-	-	-	-	-	-	-	-
A3	17.9	14.5	-	-	-	-	-	-	-	-	-	-	-	-	-	-	-	-	-
A4	18.5	19.0	19.1	-	-	-	-	-	-	-	-	-	-	-	-	-	-	-	-
A5	19.2	17.4	15.0	17.8	-	-	-	-	-	-	-	-	-	-	-	-	-	-	-
A8	19.4	19.4	18.0	19.7	19.8	-	-	-	-	-	-	-	-	-	-	-	-	-	-
A9	18.7	17.1	14.6	18.8	16.7	16.9	-	-	-	-	-	-	-	-	-	-	-	-	-
A11	19.0	17.3	17.6	22.2	19.2	19.0	16.4	-	-	-	-	-	-	-	-	-	-	-	-
A12	20.4	15.0	13.0	19.5	15.9	19.0	18.3	16.7	-	-	-	-	-	-	-	-	-	-	-
A13	18.3	11.6	14.4	17.0	16.0	19.6	18.8	16.2	15.1	-	-	-	-	-	-	-	-	-	-
A16	21.2	17.7	20.6	18.3	18.7	22.3	22.4	21.3	19.3	17.7	-	-	-	-	-	-	-	-	-
A17	16.6	13.6	10.9	17.0	13.8	18.0	13.7	18.1	13.3	15.2	17.4	-	-	-	-	-	-	-	-
A18	21.6	13.5	17.1	20.2	18.9	22.6	19.9	20.1	16.7	14.0	18.7	16.1	-	-	-	-	-	-	-
A19	18.6	15.4	13.4	17.2	14.9	16.2	5.1	15.3	16.7	16.5	20.1	12.6	17.8	-	-	-	-	-	-
A20	21.9	16.3	17.2	19.3	16.9	20.6	18.9	17.8	17.1	16.9	18.6	18.0	18.5	16.8	-	-	-	-	-
A21	18.5	16.3	16.1	21.9	17.0	21.5	19.0	17.0	16.5	15.4	19.7	17.4	17.2	18.2	19.3	-	-	-	-
A22	28.0	23.9	22.5	27.3	22.1	27.1	25.7	26.6	24.9	25.8	27.5	23.6	23.6	25.3	23.6	23.9	-	-	-
A23	18.3	13.1	15.4	17.8	16.7	18.1	15.9	18.6	14.9	13.5	17.5	15.0	13.9	14.7	13.6	17.7	24.6	-	-
A24	19.0	12.5	15.8	19.1	15.6	18.4	18.0	17.1	15.3	12.9	18.4	15.5	12.6	17.2	14.7	15.4	22.3	12.2	-

## Data Availability

All data generated and analyzed in this study are included in this article.

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
