# Peer review of "Molecular Characterization of Small Ruminant Lentiviruses in Polish Mixed Flocks Supports Evidence of Cross Species Transmission, Dual Infection, a Recombination Event, and Reveals the Existence of New Subtypes within Group A"

_viruses, 2021, doi:10.3390/v13122529_

Round 1

Reviewer 1 Report

The manuscript by Olech and Kuzmak describes the genomic study of a large number of sheep and goats in different parts of Poland in order to classify them in subtypes. They find some evidence of interspecies transmission as well as recombination. These results are important and illustrate both these facts, that are assumed by many researchers but few times demonstrated. In general it is well written and easy to follow.

Major point:

Other researchers have found that not all seropositive animals are PCR positive. However, the authors have been successful in amplifying SRLV genomes from all seropositive animals. This makes me wonder whether some of the PCR results may be false positives. Convincing evidence should be supplied as to this matter. Also, seronegative animals may be PCR positive. As it would also have been interesting to study this fact as the genomes from these animals might have been different, some studies should be performed including seronegative PCR-positive animals.

Minos points:

There are many more transcription factor binding sites than those analyzed in the study and the selection is not well substantiated.

The authors mention that there is antigenic variation in the virus and imply that this limits the effectiveness of serological methods. However, the study does not prove this statement and thus, this sentence (L83-85) should be removed.

The primers used for classifying the strains into the different subtypes should be better described, as they are a fundamental part of the study. Also, please include the sizes of the amplicons.

The differences between the genomes of VMV and CAEV should be emphasized.

Since the only part of the LTR that contains promoters is the U3, L58 should be changed.

The part of the Results section dealing with the phylogenetic analysis should be shortened as it presents many results shown in the figures.

Table 1. Please, indicate what do the asterisk and the dashes mean and indicate whether the Accession Number is from GenBank of from another genome repository.

Figures 1 and 2. Please, for clarity purposes, use different symbols for marking sheep and goat and indicate what the red and blue lines mean. Please, explain the roman numerals in A12. In addition, Figure 5 represents the different subtypes in different colors and this seems to be easier to follow. If possible, do it with figures 1 and 2.

Figure 3. Please, indicate what the different colors (light grey, dark grey, white, pink, blue line and yellow line) represent.

Figure 4 and 5. Please, indicate in the legend which are the reference strains and for what are they reference, and check that the same reference strains in both figures. Use the same color code for differentiating subtypes in both figures.

Figure 6. The legend does not reflect the content of the figure. It only shows part of U3, and it includes part of U5. The grey box is difficult to see.

L219-220 Please, complete the sentence indicating to what these subtypes have similarity.

The discussion is very interesting and well written. However, the paper by Gomez-Lucia (Vet J 2013) should be consulted for a better description of the TBS.

Please, include the GenBank Accession Numbers for the reference strains used.

The English language should be reviewed for better understanding (eg. L62 linear epitopes that induces, L19 were instead of have been).

Author Response

We would like thank the reviewer for his comments on our manuscript. We have acted upon the suggestions provided by the reviewer  and alterations were included in the updated version of the manuscript.

Rewiever 1

The manuscript by Olech and Kuzmak describes the genomic study of a large number of sheep and goats in different parts of Poland in order to classify them in subtypes. They find some evidence of interspecies transmission as well as recombination. These results are important and illustrate both these facts, that are assumed by many researchers but few times demonstrated. In general it is well written and easy to follow.

Major point:

Other researchers have found that not all seropositive animals are PCR positive. However, the authors have been successful in amplifying SRLV genomes from all seropositive animals. This makes me wonder whether some of the PCR results may be false positives. Convincing evidence should be supplied as to this matter. Also, seronegative animals may be PCR positive. As it would also have been interesting to study this fact as the genomes from these animals might have been different, some studies should be performed including seronegative PCR-positive animals.

Re: A total of 263 samples were investigated in this study. Out of 263 serum samples, 84 (32.0%) were positive in the ELISA test. Proviral DNA of 54 (not 84) samples originated from 26 sheep and 28 goats from six different flocks was successfully amplified and sequenced. We agree that it would be interesting to study the speciment from  seronegative animals by PCR but our  goal was to focus  on molecular analysis of proviruses. In many previous study we realized that  PCR amplification of speciment from serologically negative animals lead to rather faint bands making results inconclusive. That was the main argument to focus on analysis of  84 seropositve animals.  However, the authors believe that the reviewer's comment should be taken into account in planning future studies on the molecular structure of proviruses in animals eluding serological detection.

Minos points:

There are many more transcription factor binding sites than those analyzed in the study and the selection is not well substantiated.

Re: We tried to amplify longer fragment of LTR containing more transcription factor binding sites but many samples were PCR-negative. In the future we would like design a new primers which allow to analyze longer fragment of LTR.

The authors mention that there is antigenic variation in the virus and imply that this limits the effectiveness of serological methods. However, the study does not prove this statement and thus, this sentence (L83-85) should be removed.

Re: This sentence does not refer to the results of our study. Results of presented  study are discussed in section „Discussion”. In introduction are included general informations based on current stage on the SRLVs research field.

The primers used for classifying the strains into the different subtypes should be better described, as they are a fundamental part of the study. Also, please include the sizes of the amplicons.

Re: It has been corrected.

The differences between the genomes of VMV and CAEV should be emphasized.

Re: Originally, MVV and CAEV were considered as distinct viral species restricted to sheep and goats, respectively, but several reports indicated that there are different lentiviral subtypes able to infect both sheep and goats. The. CAEV and MVV share many features, and they are both considered to be SRLV.

Since the only part of the LTR that contains promoters is the U3, L58 should be changed.

Re: It has been corrected.

The part of the Results section dealing with the phylogenetic analysis should be shortened as it presents many results shown in the figures.

Re: According to the authors, the data included in the figures 1 and 2 reflected some specific aspect of phylogenetic analysis and therefore, it should be sufficiently described so that everyone can understand what is shown.  That is why the authors are convinced that a full description should be included in the text.  

Table 1. Please, indicate what do the asterisk and the dashes mean and indicate whether the Accession Number is from GenBank of from another genome repository.

Re: It has been corrected.

Figures 1 and 2. Please, for clarity purposes, use different symbols for marking sheep and goat and indicate what the red and blue lines mean. Please, explain the roman numerals in A12. In addition, Figure 5 represents the different subtypes in different colors and this seems to be easier to follow. If possible, do it with figures 1 and 2.

Re: Figures had been corrected. In Figures 1 and 2 we introduced  different  symbols for marking sheep and goats. Sequences from this study are labeled by black circles and their names are proceeded by the flock origin and the animal species (s-sheep; g-goat).

Figure 3. Please, indicate what the different colors (light grey, dark grey, white, pink, blue line and yellow line) represent.

Re: It has been corrected.

Figure 4 and 5. Please, indicate in the legend which are the reference strains and for what are they reference, and check that the same reference strains in both figures. Use the same color code for differentiating subtypes in both figures.

Re: It has been corrected.

Figure 6. The legend does not reflect the content of the figure. It only shows part of U3, and it includes part of U5. The grey box is difficult to see.

Re: It has been corrected.

L219-220 Please, complete the sentence indicating to what these subtypes have similarity.

Re: The indicated sentence informs about the similarity of sequences forming the subtype A23 and A24 (intra-subtype similarity). It has been corrected.

The discussion is very interesting and well written. However, the paper by Gomez-Lucia (Vet J 2013) should be consulted for a better description of the TBS.

Re: The LTR fragment analysed in Gomez-Lucia (Vet J 2013) paper is longer than LTR fragment analysed in our study so it was impossible to analyse/describe/compare all transcription factor binding sites included in Gomez-Lucia paper with those described in our study. However, we the Gomez-Lucia (Vet J 2013) paper has been included in the Discussion.

Please, include the GenBank Accession Numbers for the reference strains used.

Re: The GenBank accession numbers of reference strains are indicated on trees and in the Figure legends.

The English language should be reviewed for better understanding (eg. L62 linear epitopes that induces, L19 were instead of have been).

Re: It has been corrected.

Reviewer 2 Report

General considerations:

The authors of this excellent manuscript extended their previous studies on SRLV circulating in Poland by analyzing samples from 17 mixed flocks of sheep and goats living in different regions of this country. This type of analysis is essential to permit a tailored serological survey applying antigens relevant to the viruses circulating in a particular epidemiological compartment.

The results obtained are solid, and their interpretation sound.

Specific criticism:

Lines 173-177: PCR was successful in 84% of the seropositive sheep, while only 53% of the seropositive goats showed a positive PCR result. Are some caprine sequences escaping detection by the primers used?

Lines 292-298: Are these two conserved breakpoints in a similar region as those previously described in Virology 487 (2016) 50–58?

Lines 358-359: The authors affirm that the sequences encoding Gag and Env proteins evolved under negative selective pressure. This could be expected for gag but is somewhat surprising for env. In fact, their discussion (lines 513-528) appears to contradict this statement. This contradiction may only be apparent. The analysis of the different domains of env may show that some regions are under purifying selection while others, such as SU4, are clearly under positive selection by the host's immune response.

Figure 5: the sequences of the SU5 variable region are highly diverse between the different subtypes but show an astonishing conservation within the same subtype. This observation appears to support the hypothesis that this region may function as a decoy antigen and, therefore, in a given subtype, evolves under negative selection (Viruses 2018, 10, 231; doi:10.3390/v10050231).

Lines 547-567: The LTR may be a determinant of virulence, and in this paragraph, the authors discuss their analysis of the LTR of these viruses. In lines 564-566, the author states that they did not observe a correlation between a previously described 11bp deletion and the virulence of these viruses. Is it correct that the sampled animals were all clinically healthy and without signs of SRLV infection? If yes, this may indicate that the LTR of these viruses may show markers of attenuation. Comparing these LTRs with those described in other attenuated SRLV isolates, e.g., Arch. Virol. (2005) 150, 201–213 and J. Gen. Virol. (2016), 97, 1699–1708, maybe worthwhile.  

Minor details:

Line 47: it may be worthwhile to mention that on rare occasions, young kids may develop a leukoencephalomyelitis with CNS signs (J. gen. Virol. 1980, 50: 69-79).

Line 55: tat should be substituted by vpr-like (J. Virol. 2003, 77: 9632–9638).

Author Response

We would like thank the reviewer for his comments on our manuscript. We have acted upon the suggestions provided by the reviewer  and alterations were included in the updated version of the manuscript.

Rewiever 2

The authors of this excellent manuscript extended their previous studies on SRLV circulating in Poland by analyzing samples from 17 mixed flocks of sheep and goats living in different regions of this country. This type of analysis is essential to permit a tailored serological survey applying antigens relevant to the viruses circulating in a particular epidemiological compartment.

The results obtained are solid, and their interpretation sound.

Specific criticism:

Lines 173-177: PCR was successful in 84% of the seropositive sheep, while only 53% of the seropositive goats showed a positive PCR result. Are some caprine sequences escaping detection by the primers used?

Re: In this study very conservative fragment of gag gene were analyzed and primers used  allowed to the amplification of  both, MVV-like and CAEV-like sequences. However, looking at the variety of subtypes circulating in Poland it is possible that primers used in this study are unable to amplify new unknown SRLVs subtypes, including those found in goats.

Lines 292-298: Are these two conserved breakpoints in a similar region as those previously described in Virology 487 (2016) 50–58?

Re: Sequence of strain 7385 was not included in the phylogenetic and recombination study, however,  additionally performed analysis (not shown in the manuscript) showed that the breakpoints are different.

Lines 358-359: The authors affirm that the sequences encoding Gag and Env proteins evolved under negative selective pressure. This could be expected for gag but is somewhat surprising for env. In fact, their discussion (lines 513-528) appears to contradict this statement. This contradiction may only be apparent. The analysis of the different domains of env may show that some regions are under purifying selection while others, such as SU4, are clearly under positive selection by the host's immune response.

Re: The results of our study showed that for both fragments, the dN/dS ratio was below 1, showing a negative selection. We definitely agree with the reviewer that some regions in env gene  may be under positive selection, what was mentioned in discussion. But such an assumption is difficult to prove because in present study  we analyzed only a short region of the env gene and the analysis of whole glycoprotein sequences would possible be more adequate.

Figure 5: the sequences of the SU5 variable region are highly diverse between the different subtypes but show an astonishing conservation within the same subtype. This observation appears to support the hypothesis that this region may function as a decoy antigen and, therefore, in a given subtype, evolves under negative selection (Viruses 2018, 10, 231; doi:10.3390/v10050231).

Re: The suggestion offered by the reviewer was included in the discussion.

Lines 547-567: The LTR may be a determinant of virulence, and in this paragraph, the authors discuss their analysis of the LTR of these viruses. In lines 564-566, the author states that they did not observe a correlation between a previously described 11bp deletion and the virulence of these viruses. Is it correct that the sampled animals were all clinically healthy and without signs of SRLV infection? If yes, this may indicate that the LTR of these viruses may show markers of attenuation. Comparing these LTRs with those described in other attenuated SRLV isolates, e.g., Arch. Virol. (2005) 150, 201–213 and J. Gen. Virol. (2016), 97, 1699–1708, maybe worthwhile.  

Re: To the authors' knowledge, all animals used in this study did not shown any clinical signs. To confirm whether a virus is attenuated, it is not enough to compare LTR sequences or even compare whole-genome sequences. To ascertain their attenuated phenotype, experimental infections should be performed to link the genetic features to a specific phenotype in goats. Such studies may be conducted in the future as they could be very interesting.

Minor details:

Line 47: it may be worthwhile to mention that on rare occasions, young kids may develop a leukoencephalomyelitis with CNS signs (J. gen. Virol. 1980, 50: 69-79).

Re: It has been included.

Line 55: tat should be substituted by vpr-like (J. Virol. 2003, 77: 9632–9638).

Re: It has been corrected.

Round 2

Reviewer 1 Report

The clarity of the manuscript has greatly improved with the addition of colors to the figures. The English language needs to be reviewed (for example, in Table 1 it should be Primer pairs instead of Primers pair). In my opinion, the manuscript should be published.